# Transverse Impact on Rectangular Metal and Reinforced Concrete Beams Taking into Account Bimodularity of the Material

**DOI:** 10.3390/ma13071579

**Published:** 2020-03-29

**Authors:** Alexey Beskopylny, Besarion Meskhi, Elena Kadomtseva, Grigory Strelnikov

**Affiliations:** 1Department of Transport Systems, Faculty of Roads and Transport Systems, Don State Technical University, Gagarin, 1, 344000 Rostov-on-Don, Russia; 2Department of Life safety and Environmental Protection, Faculty of Life Safety and Environmental Engineering, Don State Technical University, Gagarin, 1, 344000 Rostov-on-Don, Russia; reception@donstu.ru; 3Department of the Strength of Materials, Faculty of Civil Engineering, Don State Technical University, Gagarin, 1, 344000 Rostov-on-Don, Russia; elkadom@yandex.ru (E.K.); pioner0236@mail.ru (G.S.)

**Keywords:** bimodulus, stress–strain state, reinforced beam, metal beam, numerical analysis

## Abstract

This article is devoted to the stress–strain state (SSS) study of metal and reinforced fiber-reinforced concrete beam under static and shock loading, depending on the bimodularity of the material, the mass of the beam, and the location of the reinforcing bars in zones under tension and compression. It is known that many materials have different tensile and compression properties, but in most cases, this is not taken into account. The calculations were carried out by using load-bearing metal beams made of silumin and steel and reinforced concrete beams under the action of a concentrated force applied in the middle of the span. The impact load is considered as the plastic action of an absolutely rigid body on the elastic system, taking into account the hypothesis of proportionality of the dynamic and static characteristics of the stress–strain state of the body. The dependences of the maximum dynamic normal stresses on the number of locations of reinforcing bars in zones under tension and compression, the bimodularity of the material, and the reduced mass of the beam are obtained. A numerical study of SSS for metal and concrete beams has shown that bimodularity allows the prediction of beam deflections and normal stresses more accurately.

## 1. Introduction

The effect of bimodularity of materials on the stress–strain state of beams, plates, and shells under the action of static and dynamic loads was studied in the works of Ambartsumyan and his colleagues [1,2,3], as well as in the works of Jones, Bert [4,5,6], and others. Although these models are often used, there are many unresolved issues related to material modeling. Several works developing classical methods are devoted to the bimodularity of the material in the study of stress–strain state (SSS) for various types of elements of engineering involving building structures (beams, plates, and shells) under the influence of static and dynamic loads. New materials used in engineering and construction also require new approaches for taking into account the heterogeneity of materials.

He et al., in [7,8,9], analytically solved the problem of bending thin plates and beams with different tensile and compression modules based on the existing simplified model. Using the conditions of the continuity for the stress components in an unknown neutral layer, we can determine the location of the neutral layer and derive the fundamental differential equation for deflection, flexural rigidity, and internal forces in a thin plate. The results show that the use of various modules in structural calculations affects the bending stiffness of a flexible thin plate and allows for a more accurate determination of SSS.

A new method for calculating the position of the neutral surface of an orthotropic layered bimodular beam was proposed in [10] by Kumar et al. Based on this original method, the bending analysis of a thick bimodular layered beam is considered, using the first-order shear strain theory for bimodular materials.

Shah et al. [11] considered the determination of deformations of a simply supported, uniformly loaded bimodule beam and the decision of the location of the neutral axis. A theoretical model of bimodular and unimodular beams was developed to calculate the maximum deviation, taking into account the displacement of the neutral axis in the case of a bimodular beam. The finite element method was used for comparison, using the concept of the Ambardzumyan bimodular model for simply supported and cantilevered bimodular and unimodular beams with concentrated load, uniformly distributed weight, and gradually changing load. It was found that the maximum deflection for a bimodular beam exceeds the maximum deviation for a unimodular beam for all types of load, which is important in the analysis of deformations of long-span structures.

The method based on the Bernoulli principle was used to calculate reinforced concrete-reinforced beams [12,13,14,15] from bimodular material. The beam was considered as statically indeterminate. The obtained solutions make it possible to calculate beams of arbitrary shape for various types of statically applied loads reinforced with an arbitrary number of bars.

The consideration of bimodularity when calculating the strength of beams, plates, and shell elements under the action of dynamic loads is critical. The following dynamic problems are considered by Benveniste [16,17,18]: (a) time-dependent harmonic, axial, and circumferential shear loading of a cylindrical cavity; (b) time-dependent normal loading of a spherical cavity. In both cases, the cavities are immersed in an infinite medium which is incompressible and has different behavior under tension and compression. Wave analytical solutions are obtained, the results of which are compared with the results of classical elastic solutions.

The flexural–vibrational behavior of bimodular layered composite cylindrical panels with various boundary conditions is considered in [19,20,21]. The formulation is based on the theory of first-order shear deformation and Bert’s constitutive model. Governing equations are obtained by using the finite element method and the Lagrange equation of motion. An iterative approach to eigenvalues is used to obtain positive and negative frequencies of free oscillations of the half-period and the corresponding modes. A detailed parametric study of the influence of the thickness ratio, aspect ratio, lamination pattern, boundary conditions, and bimodularity coefficient on the free vibration characteristics of bimodular angular and transverse layered composite cylindrical panels was carried out. It is interesting to note that there is a significant difference between the frequencies of positive and negative half-cycles, depending on the panel parameters. The distribution of modal stresses in thickness for the positive half-cycle is significantly different from that for the negative half-period, in contrast to the unimodular case when the stresses at a certain place in the negative half-period would be of the same magnitude but of opposite sign corresponding to the positive half-period. Finally, for a typical case, the effect of bimodularity on the stationary characteristic is studied in comparison with the frequency ratio of forcing. Stresses under dynamic loading are different for the positive and negative half of the vibration cycle.

Many graded materials have different tensile and compression moduli. One-dimensional and two-dimensional mechanical models of a functionally graded beam with a bimodular effect were established for the first time. A material that not only possesses a functionally graded characteristic, but also demonstrates various tensile and compression elastic moduli, is considered in [22]. Analytical solutions of a bimodular functionally graded beam with pure bending and bending in the transverse direction were obtained, following the gradient function as an exponential expression. It was shown that, due to the introduction of a dual-module functional gradient effect of materials, maximum tensile and compressive bending stresses might not occur in the lower and upper parts of the beam.

A lot of materials demonstrate bimodularity, which is critical in electronics, medicine, engineering, and other industries. Pastor-Artigues et al. determined the mechanical properties of polylactic acid (PLA) under tensile, compressive, and bending stresses [23]. The finite element model is used to verify differences in tensile and compression characteristics, including geometric non-linearity for realistic reproduction of conditions during physical tests. It is shown that the currently used test methods do not guarantee a consistent set of mechanical properties useful for numerical modeling, emphasizing the need to identify new characterization methods that are better adapted to PLA behavior. Experiments show that PLA has double asymmetry in the behavior of tension and compression, indicating the need to process this material by using a bimodular model of elasticity.

Thus, it can be seen that many materials, including metals, alloys, concrete, organic fabrics, and others, have different properties of stretching and compression. However, this is not taken into account in practical calculations. Moreover, this review shows that the bimodality of materials can have a critical impact on strength calculations, and therefore on the size and durability of structures. Dynamic effects and inertial forces, in combination with the bimodality, significantly change the SSS of structure. Thus, the purpose of this work is to develop a simplified engineering method for analyzing the stress–strain state of bimodular material structure elements under the action of shock loads.

## 2. Materials and Methods 

Let us consider the behavior of metal beams under the action of static and dynamic loads, both with and without bimodularity. For metals, the tensile and compression moduli do not differ much. Therefore, for steel 40 (C 0.37%–0.45%, Si 0.17%–0.37%, Mn 0.5%–0.8%, Ni 0.25%, Cr 0.25%), the elastic modulus of compression $E_{c}$ = 216,110 MPa, tensile strength $E_{t}$ = 209,990 MPa, for silumin $E_{c}$ = 74,920 MPa, $E_{t}$ = 209,990 MPa [24]. The data are presented by Ambardzumyan, according to the results of tests of various materials for uniaxial tension and compression.

The effect of the bimodularity of the material was confirmed by comparing the theoretical value of the maximum deflection ($f_{s}$) with the experimental ($f_{se}$).

In this work, the values of the maximum deflection of a simply supported rectangular beam 4 × 20 mm are determined under the action of a concentrated force of 10 N, applied in the middle of the span. The experimental study of static deflection was carried out on the device shown in Figure 1.

During the experiment, a three-point bend of a 4 × 20 mm beam was loaded, and the deflection was measured at the point of application of force (Figure 1) hour-type indicator. The results are presented in Table 1 and Table 2.

We tested articulated beams of rectangular cross-section 4 × 20 mm, under the action of a statically applied concentrated force in the middle of the span. Beam materials were silumin and steel. The deflection in the middle of the beam span was determined. The conducted experiment on metal beams showed a good agreement of the deflection value of the experimental data and the calculated value with the bimodularity of the material. Taking into account the bimodularity allows you to more accurately determine the deflections that coincide with the experiment in statics. The static deflection value is included in the dynamic coefficient formula for determining the maximum normal stresses. A more precise determination of static deflection allows us to determine the dynamic maximum normal stresses more accurately.

In this paper, we study the SSS of a beam made of structural fiber-reinforced concrete. Polyamide fibers are used as fibers. Fiber–concrete as structural or insulating foam concrete (GOST 25485-89) reinforced with fiber (GOST 14613–83) was made in accordance with Russian standards. The use of this material in construction makes it possible to several times lower the heat loss of buildings. For example, the walls of such products prevent significant heat leakage in the winter and protect the indoor climate from excessively high temperatures in the summer. To reduce shrinkage deformations and improve the uniformity of the structure of porous concrete, reinforcing fibers are added to the composition of the mixture for preparing the material. Such filling significantly improves the physical and mechanical properties of finished products. For example, with the addition of polypropylene fiber in an amount of 0.4% of the total cement volume, an increase in the compressive strength of the D400 foam concrete grade is increased to 26%.

The considered type of foam concrete by its functional purpose is divided into three independent groups:Heat-insulating, density 400–500 kg/m^3^.Structurally insulating, 600–1100 kg/m^3^.Structural, density 1100–1200 kg/m^3^.

For a steel beam, the theoretically obtained deflection of the beam, taking into account bimodularity, is 4% more than the experimental one and excluding bimodularity by 5.3%. For a silumin beam, the theoretical deflection is 0.7% more than the experimental one, taking into account bimodularity, and 5.4% without, taking into account bimodularity. The results obtained confirm the need to take into account the bimodularity property when calculating the strength and stiffness of structural elements.

In the calculations of concrete structures, fiber-reinforced concrete with a tensile–compression modulus of 5000 MPa was used, if the material is considered as isotropic. Finding the heterogeneous properties of concrete, various elastic moduli under the tension of 5000 MPa and compression of 2550 MPa are accepted in the calculations.

## 3. A Heterogeneous (Bimodule) Model of a Beam

A heterogeneous (bimodule) model of a reinforced beam under the action of static loads is presented as a beam consisting of two layers: stretched, compressed, and reinforcement bars. Considering the beam as statically indeterminable, we obtain the equilibrium equation for a heterogeneous beam,
(1)$$M_{y}=M_{yt}+M_{yc}+M_{ya}\;$$
and condition for compatibility of deformations of a heterogeneous beam,
(2)$$\frac{1}{\rho }=\frac{1}{\rho_{t}}=\frac{1}{\rho_{c}}=\frac{1}{\rho_{a}}\;$$
where $M_{y},\;\frac{1}{\rho }$ are the bending moment and curvature of the beam; $M_{yt},\;\frac{1}{\rho_{t}}$ are the bending moment and curvature of the beam of the zone under tension; $M_{yc},\;\frac{1}{\rho_{c}}$ are the bending moment and curvature of the beam of the zone under compression; and $M_{ya},\;\frac{1}{\rho_{a}}$ are the bending moment and curvature of reinforced bars.

The equilibrium condition (1) ∑My = 0 expressed in terms of normal stresses has the following form:(3)$$M_{y}=\int_{A}\sigma \;z\;dA=\int_{A_{t}}\sigma_{t}\;z\;dA\;+\;\int_{A_{c}}\sigma_{c}\;z\;dA\;+\;\int_{A_{a}}\sigma_{a}\;z\;dA=M_{yt}+M_{yc}+M_{ya}$$
where $\sigma_{t},\;A_{t}$—normal stress and cross-sectional area of the beam of the stretched zone; $\sigma_{c},\;A_{c}$—normal stress and cross-sectional area of the beam in the compression zone; and $\sigma_{a},\;A_{a}$—normal stress and cross-sectional area of reinforcement bars.

By substituting normal stresses $\sigma_{t}=\frac{E_{t\;}z}{\rho },\sigma_{c}=\frac{E_{c}z}{\rho },\sigma_{a}=\frac{E_{a}z}{\rho }\;$ into (3), we obtain the neutral line curvature formula for a beam of bimodular material:(4)$$M_{y}=\frac{1}{\rho }\left(E_{t}J_{y}^{t}+E_{c}J_{y}^{c}+E_{a}\left[n_{t}\left(J_{y_{1}}^{t}+A_{a}^{p}c_{t}^{2}\right)+n_{c}\left(J_{y_{1}}^{c}+A_{a}^{c}c_{c}^{2}\right)\right]\right)\;$$

For a beam, we have the general curvature formula:(5)$$\frac{1}{\rho }=\frac{M_{y}}{D}=\frac{M_{y_{p}}+M_{y_{c}}+M_{y_{a}}}{D},\;$$
where *D* is the reduced stiffness of the beam of bimodular material; $E_{t}$ is the modulus of elasticity of the material in the tensile zone; $E_{c}$ is the modulus of elasticity of the material in the compression zone; $E_{a}$ is the modulus of elasticity of the reinforcement; $J_{y}^{t}$ is the moment of inertia of that part of the cross section that lies in the stretch zone relative to the neutral axis; $J_{y}^{c}$ is the moment of inertia of that part of the cross-section, which lies in the compression zone, relative to the neutral axis; $J_{y_{1}}^{t}$ is a moment of inertia of the cross section of the reinforcement, which lies in the tension zone, relative to its own central axis; $J_{y_{1}}^{c}$ is the moment of inertia of the cross section of the reinforcement, which lies in the compression zone, relative to its own central axis; $n_{t}$ is the number of reinforcement bars in the tension zone; $n_{c}$ is the number of reinforcement bars in the compression zone; $A_{a}^{t}$ is the cross-sectional area of the reinforcement in the tension zone, $A_{a}^{c}$ is the cross-sectional area of the reinforcement in the compression zone; $c_{t}$ is the distance from the reinforcement in the tension zone to the neutral axis; and $c_{c}$ is the distance (coordinate) from bars in the compression zone to the neutral axis.

From Equations (4) and (5), we obtain the reduced-stiffness expression for reinforced beams of heterogeneous material *D*:(6)$$D=E_{t}J_{y}^{t}+E_{c}J_{y}^{c}+E_{a}\left[n_{t}\left(J_{y_{1}}^{t}+A_{a}^{p}c_{t}^{2}\right)+n_{c}\left(J_{y_{1}}^{c}+A_{a}^{c}c_{c}^{2}\right)\right]\;$$

To determine the position of the neutral line, we consider another static equation—the projection onto the axis of the bar, $\sum F_{x}\;=\;0$:(7)$$\underset{A}{\int }\sigma dA=\underset{A_{p}}{\int }\sigma_{t}dA+\underset{A_{c}}{\int }\sigma_{c}dA+\underset{A_{a}}{\int }\sigma_{a}dA=0\;$$

By substituting $\sigma_{t,\;}\;\sigma_{c,}\;\sigma_{a}$ in Equation (7), we obtain the following:(8)$$E_{t}S_{y}^{t}+E_{c}S_{y}^{c}+E_{a}(n_{t}A_{a}^{t}c_{t}+n_{c}A_{a}^{c}c_{c})=0\;$$
where $S_{y}^{t}$ is the static moment of that part of the cross-section that lies in the tension zone, relative to the neutral axis; and $S_{y}^{c}$ is the static moment of that part of the cross-section that lies in the compression zone, relative to the neutral axis.

Normal stress formulas taking into account Equations (4) and (5) have the following form:(9)$$\sigma_{t}=\frac{E_{p}M_{y}}{D}z,\sigma_{c}=\frac{E_{c}M_{y}}{D}z,\sigma_{a}=\frac{E_{a}M_{y}}{D}z.$$

For a rectangular cross-section, Equation (9) for the maximum normal tensile stress $\sigma_{t}$ and the maximum normal compressive stress $\sigma_{c}$, taking into account the bimodularity of the material for reinforced beams during bending under static loads, has the following form [11]:(10)$$\left|\sigma_{\max t}\right|={\frac{3(1+\sqrt{k})\sqrt{k}h}{kbh^{3}+3{\left(1+\sqrt{k}\right)}^{2}E_{a}(n_{t}I_{a+}+n_{c}I_{a-})/E}}_{t}\left|M_{\max y}\right|$$
(11)$$\left|\sigma_{\max c}\right|=\frac{3k(1+\sqrt{k})h}{kbh^{3}+3{\left(1+\sqrt{k}\right)}^{2}E_{a}(n_{t}I_{a+}+n_{c}I_{a-})/E_{t}}\left|M_{\max y}\right|$$
where *h* is the height of the beam; *b* is the width of the beam; $M_{y}$ is the bending moment relative to the neutral line in an arbitrary cross-section of the beam; $E_{a}$ is the tensile modulus of reinforcement bars; $E_{t}$ is the modulus of elasticity of concrete (aggregate) in tension; $I_{a+}$ is the axial moment of inertia of the cross-section of one reinforcement bar in the tensile zone; $E_{c}$ is the modulus of elasticity of concrete (aggregate) in compression; $I_{a-}$ is the axial moment of inertia of the cross-section of one reinforcement bar in the compression zone; $n_{c}$ is the number of bars in the compression zone; $n_{t}$ is the number of bars in the tension zone; and $k=\frac{E_{c}}{E_{t}}$.

The stressed state of the beam was investigated under the action of concentrated force applied in the middle of the span of articulated metal and reinforced concrete beams. The impact is considered as an absolutely plastic impact of an absolutely rigid body on the elastic system, taking into account the hypothesis of proportionality of the dynamic and static characteristics of the stress–strain state of the body. The dynamic coefficient ($k_{d}$) [25] is determined to take into account and without taking into account the bimodularity of the beam material.

The formulas determine the dynamic coefficient.

Excluding beam mass:(12)$$k_{d}=1+\sqrt{1+\frac{2h}{f_{s}}}$$

Considering the mass of the beam:(13)$$k_{d}=1+\sqrt{1+\frac{2h}{f_{s}}{\left(1+\frac{M_{B}}{M_{A}}\right)}^{-3}}$$
where *h* is the height of the load; $f_{s}$ is the static beam deflection under load, without taking into account the mass of the beam; $M_{A}$ is the mass of the falling load; $M_{B}$ is the reduced beam mass, according to Cox [26].

For a simply supported beam, loaded in the middle of the span *L*, $M_{B}=\frac{17}{35}\;m_{B}\;L$, where $m_{B}$ is the distributed mass of the beam, and *L* is the length of the beam.

As can be seen in Table 3, the dynamic coefficient decreases with the increasing ratio $m_{B}L/M_{A}$, and $m_{B}L/M_{A}\;>10$
$k_{d}$ = 2. Therefore, the initial data were taken in this study, ensuring the strength and rigidity of the beam and allowing us to study in enough detail the effect of various parameters on impact strength.

## 4. Results

### 4.1. Determination of Dynamic Stresses Arising in a 4 × 20 Metal Rectangular Beam under the Influence of a Falling Load of Mass M_A_ from a Height (h) to the Middle of the Beam

When determining the maximum stresses in metal beams, put $n_{c}$ = $n_{t}$ = 0 in Equations (10) and (11). Table 4 shows the maximum stresses, $\sigma_{\max t}$ and $\sigma_{\max c}$, for the mass of the falling load, $M_{A}$ = 1.00 kg, and the height of the falling load, *h* = 10.00 mm.

Table 5 shows the maximum stresses ($\sigma_{\max t}$ and $\sigma_{\max c}$) for the mass of the falling load, *M_A_* = 0.10 kg, and the height of the falling load, *h* = 4.00 mm.

As can be seen from the results, if the beam mass is not taken into account, then the dynamic coefficient with bimodularity differs from the dynamic coefficient without bimodularity for steel by 0.5%, and for silumin by 1.8%. If the beam mass is taken into account, this leads to a difference in $k_{d}$ with and without taking into account the bimodularity of the material for steel by 0.5%, and for silumin by 1.5%.

The difference between the dynamic coefficients excluding bimodularity, taking into account and excluding the mass of the beam, for steel is 16%, and for silumin is 56%. The difference between the dynamic coefficients, taking into account bimodularity, taking into account and without taking into account the mass of the beam, is 16% for steel and 57% for silumin.

From the results obtained, it can be concluded that, for metal beams, the dynamic coefficient has a greater influence on taking into account the mass of the beam than considering the bimodularity of the material.

### 4.2. Determination of Dynamic Stresses in a Simply Supported Reinforced Concrete Rectangular Beam under the Influence of a Falling Load of Mass ($M_{A}$) from a Height (h) to the Middle of the Span

The material of the beam is fiber-reinforced concrete with elastic moduli for compression $E_{c}$ = 2250 MPa, and tensile $E_{t}$ = 5000 MPa. We consider structural fiber-reinforced concrete with polyamide fibers made by standard GOST 25485-89 [27]. 

#### 4.2.1. Investigation of the Influence of the Location of the Reinforcement on the Dynamic Coefficient and Maximum Normal Stresses Arising in the Cross-Section, Taking into Account the Beam Mass and Bimodularity of the Material

Beam parameters: beam length *L* = 4.0 m; cross-sectional dimensions *h* = 89.0 cm, *b* = 28 cm, $d_{t}$ = 0.012 m is the diameter of the reinforcement bar in the tension zone, and $d_{c}$ = 0.008 m is the diameter of the reinforcement in the compression zone. Table 6 shows the maximum dynamic normal stresses when the bars are located only in the stretched zone $n_{t}$ = 4, $n_{c}$ = 0. The mass of the falling load $M_{A}$ = 100 kg. The reduced beam mass is $M_{B}$ = 338.9 kg. Drop height is *h* = 40 mm. $m_{B}L/M_{A}$ = 3.3891, 2*h*/$f_{s}$ = 5156. 

Table 7 shows the maximum dynamic normal stresses at the location of the same number of bars located in the compressed and in the stretched zone. The mass of the falling load is $M_{A}$ = 100 kg. Drop height is *h* = 40.00 mm, $n_{t}$ = 4, $n_{c}$ = 4.

The obtained calculations (Table 6 and Table 7) show that taking into account the bimodularity of the material reduces the dynamic coefficient by 16%, and taking into account the mass of the beam reduces $k_{d}$ by 723%.

The dynamic coefficient for the location of reinforcing bars in both the stretched and compressed zones is more than the dynamic coefficient for the location of reinforcing bars only in the extended zone by 8%.

#### 4.2.2. Investigation of the Influence of the Number of Bars Located in the Zone under Tension on the Stress state of the Reinforced Beam with and without Bimodularity, with and without Considering the Beam Mass under the Impact of External Loads

Let us consider the stress–strain state of a fiber–concrete beam, without the bimodularity of the material and without taking into account the beam mass. 

Dynamic stresses without bimodularity depending on the number of bars in the zone under tension, without considering the beam mass at $n_{c}$ = 0, $n_{c}$ = 2, $n_{c}$ = 4, are shown in Figure 2. $E_{c}$ = 5000 MPa, $E_{t}$ = 5000 MPa.

An increase in the number of bars ($n_{t}$) in the zone under tension significantly reduces the maximum dynamic tensile stresses. Adding one bar in the stretched zone reduces the dynamic maximum tensile stress by 4%, while the maximum dynamic compressive stress decreases slightly by only 1.1% (Figure 2, graph 1 and 4).

Adding two bars in the zone under compression reduces the dynamic maximum normal compressive stress by 11% and tensile by only 0.5%. It can be seen from Figure 2 that it is possible to equalize the maximum dynamic normal tensile and compressive stresses by placing two reinforcing bars in the zone under compression and one in the tensile one (Figure 2, graph 2 and 5).

An increase in the bars in the zone under compression halves the maximum dynamic normal compressive stress by 3% and tensile by only 0.1%. The maximum dynamic normal tensile and compressive stresses are equal in absolute value if there are four reinforcing bars in the zone under compression and two bars in the zone under tension (Figure 2, graph 3 and 6)

Now, let us consider the stress–strain state of a fiber concrete beam, with the bimodularity of the material and without taking into account the beam mass.

Dynamic stresses with bimodularity effect, depending on the number of bars in the zone under tension, without taking into account the beam mass at $n_{c}$ = 0, are shown in Figure 3. $E_{c}$ = 2250 MPa, $E_{t}$ = 5000 MPa. 

Figure 2 and Figure 3 show that the dependence of the maximum normal compressive and tensile stresses on the number of bars in a compressed and stretched zone is almost the same, both with and without bimodularity.

Accounting for bimodularity (Figure 2 and Figure 3) almost does not affect the value of maximum tensile stresses in the absence of reinforcement, but reduces the maximum tensile stress by 4%, with an increase in the number of reinforcement bars in the tensile zone in comparison with the value of maximum tensile stress, without considering bimodularity.

The maximum compressive normal stresses are reduced by taking into account bimodularity by 45% at $E_{t}=2E_{c}$.

As can be seen from the graphs presented in Figure 2 and Figure 3, the bimodularity of the material qualitatively changes the dependence of the maximum dynamic normal stresses on the number and location of reinforcing bars. At $E_{t}=2E_{c}$, the maximum dynamic tensile normal stresses decrease by only 1%, while the maximum dynamic compressive normal stresses decrease by 64%.

Let us consider the stress–strain state of a fiber–concrete beam, without the bimodularity of the material and by taking into account the beam mass. The dynamic stresses for this case are shown in Figure 4.

By comparing Figure 2, Figure 3 and Figure 4, it is seen that the inertial effects of the mass of the beam significantly reduce stress in zones under tension and compression. The nature of the stresses is the same.

The dependence of the maximum dynamic normal stresses on the number of reinforcing bars in a compressed and stretched zone, taking into account that the mass of the beam is the same as without the mass of the beam (Figure 2, Figure 3 and Figure 4), but the value of the maximum dynamic normal tensile stress decreases by 727%, and the magnitude of the maximum dynamic normal compressive stress by 767%.

Now, let us consider the stress–strain state of a fiber concrete beam, with the bimodularity of the material and by taking into account the beam mass. Dynamic stresses with bimodularity effect depending on the number of bars in the zone under tension, taking into account the beam mass at $n_{c}$ = 0, as shown in Figure 5. $E_{c}$ = 2250 MPa, $E_{t}$ = 5000 MPa. 

The effect of bimodularity during a bending impact on a massive beam (Figure 5) increases the value of maximum tensile normal stresses by 2.5%, and the maximum compressive normal stresses decrease by 4.7%.

It is interesting to analyze the influence of the location of reinforcing bars with and without taking into account bimodularity and beam mass on the values of maximum normal stresses. By examining graphs 2 and 5, we see that, by increasing the number of reinforcing bars twice in the zone under tension, the maximum normal tensile stresses (Figure 2, graph 5) and the maximum normal compressive stresses (Figure 2, graph 2) decrease by 8% and 1%, respectively (excluding modularity and mass). Under the same conditions for the location of reinforcing bars, but taking into account the bimodularity of the material and the beam mass, the maximum normal tensile stresses (Figure 5, graph 5) and the maximum normal compressive stresses (Figure 5, graph 2) decrease by 16% and 1.4%, respectively.

As can be seen (or by analyzing) from graphs 2–5 (Figure 5), considering the bimodularity of the material and taking into account the mass of the beam affects the maximum normal stresses under the action of a bending shock. The bimodularity of the material in comparison with the mass of the beam has a smaller effect on the values of the maximum normal stresses at the specified physical and geometric parameters.

## 5. Conclusions

The method proposed in this work makes it possible to consider reinforced beam structural elements under the action of shock loads made of bimodular material. When comparing the experimental values of the deflections with the theoretical ones, it is evident (Table 1 and Table 2) that taking into account bimodularity gives a more accurate value of the deflection.

The value of the dynamic coefficient for metal beams is practically independent of bimodularity, since the elastic moduli under tension and compression differ little for metals. Taking into account the mass of the metal beam, the more the difference between the tensile and compression modules (Table 4 and Table 5), the more the dynamic coefficient and dynamic normal stresses decrease.

The obtained calculations (Table 5 and Table 6) show that taking into account the bimodularity of the material with given loads and mechanical characteristics of concrete affects the value of the dynamic coefficient by 45 times less than taking into account the mass of the beam compared to taking into account the bimodularity of the material.

The dynamic coefficient for the location of reinforcing bars in both the stretched and compressed zones is greater than the dynamic coefficient for the location of reinforcing bars only in the extended zone by 8%.

The dependence of the maximum tensile and normal compressive stresses on the number of reinforcing bars located in the compressed and elongated ones showed a qualitative and quantitative difference between the graphs shown in Figure 2 and Figure 3, with and without taking into account the bimodularity of the beam material. When taking into account the mass of the beam (Figure 4 and Figure 5), dynamic stresses decrease on average by 700%.

The study conducted in this work shows that, for reinforced bimodular beams’ calculation under impact load, the stress state depends on many factors: The ratio of the mass of the beam and the mass of impact load;The ratio of the height with which the load falls and the magnitude of the static deflection under the load;The ratio of tensile and compression moduli;The location of reinforcing bars in a compressed and stretched zone.

The method proposed in this work makes it possible to analyze in detail the influence of all the above factors for arbitrarily supported beams, with a cross-section of various rectangular shapes, with different mechanical characteristics of the material, and with different locations of the reinforcing bars of the beams under impact loads.

## Figures and Tables

**Figure 1 materials-13-01579-f001:**
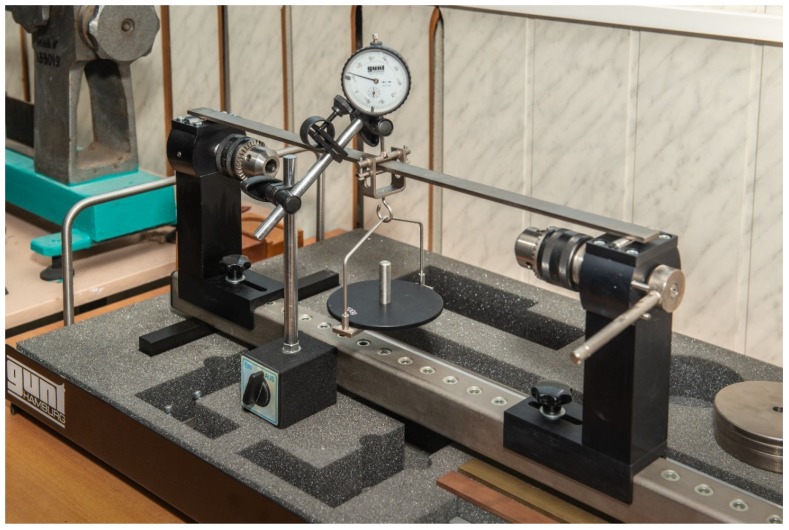
A device for the experimental study of static deformations and displacements of metal beams (G.U.N.T. Gerätebau GmbH, Hamburg, Germany).

**Figure 2 materials-13-01579-f002:**
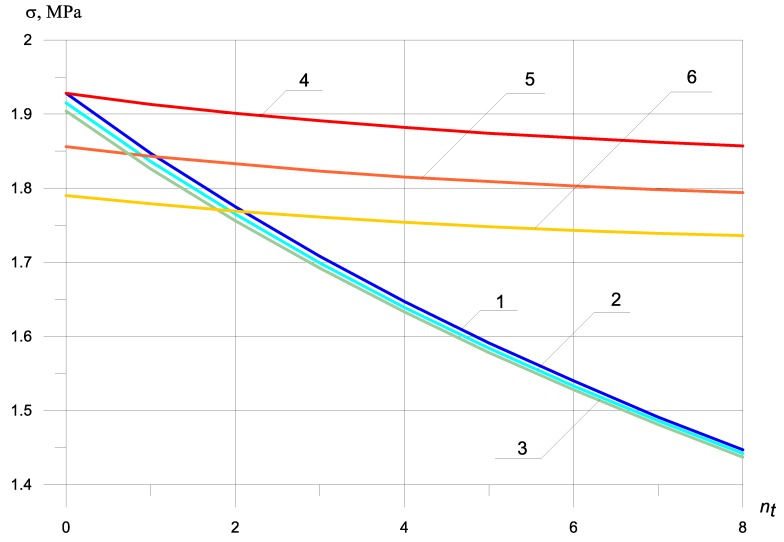
Dynamic stresses excluding bimodularity depending on the number of bars in the zone under tension, excluding the beam mass: (1) $\sigma_{t}$ at $n_{c}$ = 0; (2) $\sigma_{t}$ at $n_{c}$ = 2; (3) $\sigma_{t}$ at $n_{c}$ = 4; (4) $\sigma_{c}$ at $n_{c}$ = 0; (5) $\sigma_{c}$ at $n_{c}$ = 2; (6) $\sigma_{c}$ at $n_{c}$ = 4.

**Figure 3 materials-13-01579-f003:**
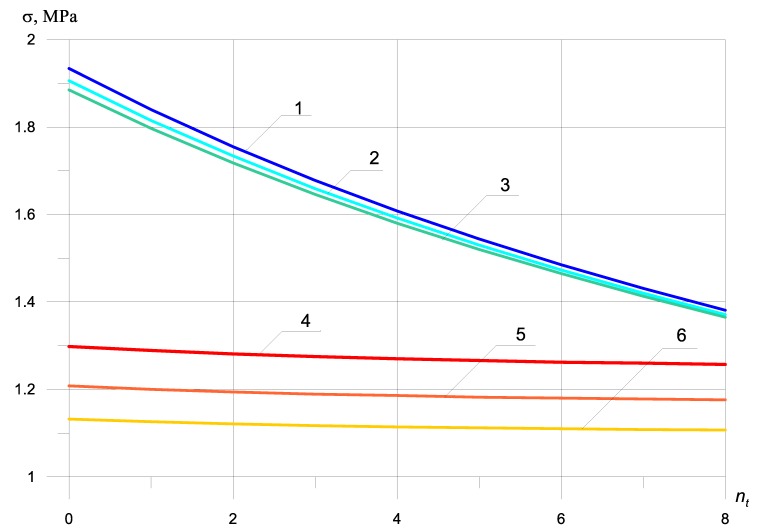
Dynamic stresses with bimodularity depending on the number of bars in the zone under tension, excluding the beam mass: (1) $\sigma_{t}$ at $n_{c}$ = 0; (2) $\sigma_{t}$ at $n_{c}$ = 2; (3) $\sigma_{t}$ at $n_{c}$ = 4; (4) $\sigma_{c}$ at $n_{c}$ = 0; (5) $\sigma_{c}$ at $n_{c}$ = 2; (6) $\sigma_{c}$ at $n_{c}$ = 4.

**Figure 4 materials-13-01579-f004:**
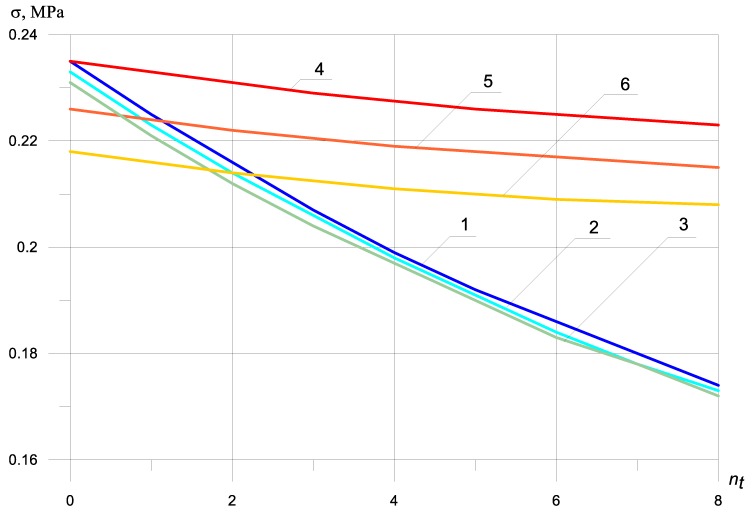
Dynamic stresses without bimodularity, depending on the number of bars in the zone under tension taking into account the beam mass: (1) $\sigma_{t}$ at $n_{c}$ = 0; (2) $\sigma_{t}$ at $n_{c}$ = 2; (3) $\sigma_{t}$ at $n_{c}$ = 4; (4) $\sigma_{c}$ at $n_{c}$ = 0; (5) $\sigma_{c}$ at $n_{c}$ = 2; (6) $\sigma_{c}$ at $n_{c}$ = 4.

**Figure 5 materials-13-01579-f005:**
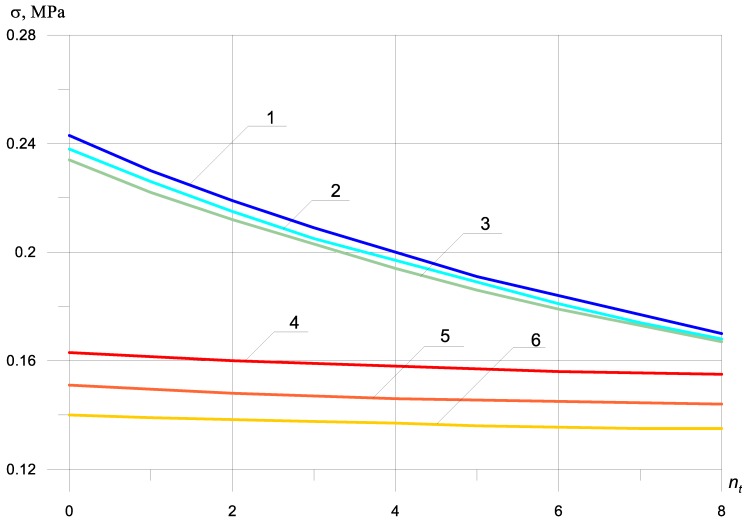
Dynamic stresses with bimodularity depending on the number of bars in the zone under tension, taking into account the beam mass: (1) $\sigma_{t}$ at $n_{c}$ = 0; (2) $\sigma_{t}$ at $n_{c}$ = 2; (3) $\sigma_{t}$ at $n_{c}$ = 4; (4) $\sigma_{c}$ at $n_{c}$ = 0; (5) $\sigma_{c}$ at $n_{c}$ = 2; (6) $\sigma_{c}$ at $n_{c}$ = 4.

**Table 1 materials-13-01579-t001:** The theoretical and experimental value of the maximum deflection of a steel beam.

Steel		
$f_{s}$, mm		$f_{se}$, mm
$E_{t}$ = $E_{c}$ = 209,990 MPa	$E_{c}$ = 216,110 MPa $E_{t}$ = 209,990 MPa	
0.953	0.940	0.900

**Table 2 materials-13-01579-t002:** The theoretical and experimental value of the maximum deflection of the beam from silumin.

Silumin		
$f_{s}$, mm		$f_{se}$, mm
$E_{t}$ = $E_{c}$ = 68,300 MPa	$E_{c}$ = 74,920 MPa $E_{t}$ = 68,300 MPa	
2.93	2.80	2.78

**Table 3 materials-13-01579-t003:** The dynamic coefficient ($k_{d}$), depending on the ratio of the load height (*h*) to the maximum deflection ($f_{s}$).

$2h/f_{s}$	$m_{B}L/M_{A}$					
	0	0.01	0.1	1.0	10	100
0	2	2	2	2	2	2
0.01	2.0050	2.0048	2.0037	2.0006	2	2
0.1	2.0488	2.0474	2.0369	2.0062	2	2
1.0	2.4142	2.4038	2.3234	2.0607	2.0004	2
10	4.3166	4.2720	3.9177	2.5000	2.0037	2
100	11.0499	10.9025	9.7253	4.6742	2.0369	2
200	15.1774	14.9685	13.2989	6.0990	2.0725	2.0001
300	18.3494	18.0932	16.0464	7.2048	2.1070	2.0001
400	21.0250	20.7291	18.3645	8.1414	2.1404	2.0002
500	23.3830	23.0521	20.4077	8.9687	2.1729	2.0002
1000	32.6386	32.1703	28.4284	12.2250	2.3234	2.0005
10,000	101.0050	99.5236	87.6842	36.3695	3.9177	2.0048

**Table 4 materials-13-01579-t004:** Maximum normal stresses ($\sigma_{\max t}$, $\sigma_{\max c}$) and dynamic coefficient ($k_{d}$) of a steel beam.

Steel	$E_{c}$ = $E_{t}$ = 209,990 MPa			$E_{t}$ = 209,990 MPa; $E_{c}$ = 216,110 MPa		
	$\sigma_{\max t},$ MPa	$\sigma_{\max c},$ MPa	$k_{d}$	$\sigma_{\max t},$ MPa	$\sigma_{\max c},$ MPa	$k_{d}$
Excluding beam mass	124.8	124.8	5.688	124.6	126.4	5.720
Given the mass of the beam	107.2	107.2	4.885	107.0	108.5	4.911

**Table 5 materials-13-01579-t005:** Maximum normal stresses ($\sigma_{\max t}$, $\sigma_{\max c}$) and dynamic coefficient ($k_{d}$) of a silumin beam.

Silumin	$E_{c}$ = $E_{t}=68,300\;\mathrm{MPa}$			$E_{t}$ = $68,300\;\mathrm{MPa};\;E_{c}=216,110\;\mathrm{MPa}$		
	$\sigma_{\max t}\;\mathrm{MPa}$	$\sigma_{\max c},\;\mathrm{MPa}$	$k_{d}$	$\sigma_{\max t}{,}_{\;}\mathrm{MPa}$	$\sigma_{\max c},\;\mathrm{MPa}$	$k_{d}$
Excluding beam mass	13.86	13.86	6.319	13.80	14.46	6.438
Given the mass of the beam	8.849	8.849	4.034	8.783	9.199	4.096

**Table 6 materials-13-01579-t006:** The maximum dynamic normal stresses when the bars are located only in the stretched zone.

Fiber Concrete	$E_{c}$ = $E_{t}=5000\;\mathrm{MPa}$			$E_{t}$ = $5000\;\mathrm{MPa};\;E_{c}=2250\;\mathrm{MPa}$		
	$\sigma_{\max t},\;\mathrm{MPa}$	$\sigma_{\max c},\;\mathrm{MPa}$	$k_{d}\;$	$\sigma_{\max t},\;\mathrm{MPa}$	$\sigma_{\max c},\;\mathrm{MPa}$	$k_{d}$
Excluding beam mass	1.647	1.647	77.652	1.608	1.270	62.276
Given the mass of the beam	0.199	0.228	9.395	0.200	0.158	7.738

**Table 7 materials-13-01579-t007:** The maximum dynamic normal stresses. The bars are located in the stretched and compressed zone.

Fiber Concrete	$E_{c}$ = $E_{t}=5000\;\mathrm{MPa}$			$E_{t}$ = $5000\;\mathrm{MPa};\;E_{c}=2250\;\mathrm{MPa}\;$		
	$\sigma_{\max t}{,}_{\;}\mathrm{MPa}$	$\sigma_{\max c},\;\mathrm{MPa}$	$k_{d}\;$	$\sigma_{\max t},$ MPa	$\sigma_{\max c},\;\mathrm{MPa}$	$k_{d}\;$
Excluding beam mass	1.633	1.754	80.835	1.580	1.114	67.827
Given the mass of the beam	0.197	0.211	9.739	0.194	0.137	8.335

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
