# Peer review of "Transverse Impact on Rectangular Metal and Reinforced Concrete Beams Taking into Account Bimodularity of the Material"

_materials, 2020, doi:10.3390/ma13071579_

Round 1

Reviewer 1 Report

Although the title of this paper is about metal and reinforced concrete beam, the reviewer is not able to any material properties or related information about concrete. The author(s) should point out clearly why this topic is appropriate for this work. The writing style is inappropriate especially in the section of Literature review in the part of Introduction, e.g. in [7-9] (line 31), in [10] (line 38), In [22] (line 77), to mention but a few. The author(s) should clearly summarize the reasons for quoting the referring researches paper and showing who is/are the author(s) of the papers for references. The inconsistency of format is found throughout the paper e.g. nc = 0 and Ec=5000 MPa (line 253). For the data presentation, some discussions are based on the comparison of different graphs, but the author showed all the graphs separately. It is impossible for the readers to observe the discussions that the author(s) made. Reviewer strongly recommends the author(s) to put the data for the comparison in the same graph in order to show what they discussed. Point form is inappropriate to be used in the conclusion. The author(s) should use number to make the conclusion clearer. Some details are mentioned as below:

  • Line 10, CCS or SSS?
  • The part of Introduction should be rewritten. Please summarize the paper cited and some important reference should give author’s names. Explain the importance of the cited paper for the current research work.
  • For the experimental study under line 107, no description for the whole experiment. Please add all the details for conducting such experiment and give the reason for its importance.
  • Line 162, additional bracket is found in the equation.
  • Line 197, in Table 3 rather than in table 3.
  • Consistent format is expected for = (spacing) throughout the paper.
  • Line 241, Table 6 and 7 rather than Table 6,7.
  • Line 270, Fig. 2 rather than fig. 2
  • Line 278, Fig. 3 and 4 rather than Fig 3,4.
  • Review all the graph and combine them for the comparison in discussion.
  • Line 340, 345, 346: see point 7
  • Line 357: see point 6.
  • No point form should be used in the conclusion.

Author Response

Response to Reviewer 1 Comments

The authors thank the reviewer for their attention to our work. We are sure that the comments made by the reviewer allowed us to make our manuscript better.

Point 1: Although the title of this paper is about metal and reinforced concrete beam, the reviewer is not able to any material properties or related information about concrete. The author(s) should point out clearly why this topic is appropriate for this work.

Response 1: We have reworked section 2 Materials and methods and specified the properties of the materials discussed in the article. As the materials, steel, silumin, and fiber concrete were selected as the most common in construction. The main goal of the work is to show how the consideration of the bimodality of materials affects the obtained values of the stress-strain state in statics and dynamics. And what discrepancies can be if we do not take into account the bimodularity.

Point 2: The writing style is inappropriate especially in the section of Literature review in the part of Introduction, e.g. in [7-9] (line 31), in [10] (line 38), In [22] (line 77), to mention but a few. The author(s) should clearly summarize the reasons for quoting the referring researches paper and showing who is/are the author(s) of the papers for references

Response 2: Thank you very much for your careful examination of our paper. We have reworked section 1 Introduction and summarized the reasons for the citation of a research paper. Following your valuable comment, we have checked many inaccuracies in the manuscript and inaccurate descriptions. We also indicated the authors of the referred works.

Point 3: The inconsistency of format is found throughout the paper e.g. nc = 0 and Ec=5000 MPa (line 253).

Response 3: The authors are grateful to the reviewer for their attentive attitude to our article. We fixed formatting throughout the text. The article has been completely revised taking into account your comments.

Point 4: For the data presentation, some discussions are based on the comparison of different graphs, but the author showed all the graphs separately. It is impossible for the readers to observe the discussions that the author(s) made. Reviewer strongly recommends the author(s) to put the data for the comparison in the same graph in order to show what they discussed.

Response 4: We reworked all the Figures and left 4 with more graphs instead of 12. This fix makes it easier to explain the identified dependencies and results to readers.

Point 5: Point form is inappropriate to be used in the conclusion. The author(s) should use number to make the conclusion clearer.

Response 5: The authors thank the reviewer for their attention to the article. We removed the Points and added numeric symbols.

Point 6: Line 10, CCS or SSS?

Response 6: We are very sorry, of course, SSS. We fixed it.

Point 7: Part of the Introduction should be rewritten. Please summarize the paper cited, and some important references should give the author’s names. Explain the importance of the cited paper for the current research work.

Response 7: The Introduction section has been significantly revised. We added the authors of the articles that we referred to. Corrections have been made for each reference in terms of its importance from the point of view of the work under consideration. The paragraph with the research goal has been significantly expanded.

Point 8: For the experimental study under line 107, no description for the whole experiment. Please add all the details for conducting such experiment and give the reason for its importance.

Response 8: Thank you for your good advice. We have added a section describing the experiment and installation for static deflection measurement. We also added the importance of the obtained experimental data in their use in dynamic calculations.

Point 9: Line 162, additional bracket is found in the equation.

Response 9: Thank you for your helpful comment. The extra bracket is removed.

Point 10: Line 197, in Table 3 rather than in table 3. Line 241, Table 6 and 7 rather than Table 6,7. Line 270, Fig. 2 rather than fig. 2. Line 278, Fig. 3 and 4 rather than Fig 3,4. Line 340, 345, 346. Line 357. Consistent format is expected for = (spacing) throughout the paper.

Response 10: Thank you very much for your comment and careful examination of our paper. All inaccuracies have been corrected. The text of the article is replaced everywhere.

Point 11: No point form should be used in conclusion.

Response 11: We removed the Points in the Conclusions and added numeric symbols.

Finally, we would like to thank you very much for your support. The paper has been carefully checked and proofread.

Reviewer 2 Report

This paper presents rather interesting results and is written properly, apart from some minor issues that are listed in what follows. Below, I also leave a set of comments to the consideration of the authors, requesting that they are duly accounted for during the revision process.

1) Lines 12-13: it is suggested to replace the expression "in the compressed and stretched zones" with "in zones under tension and compression". Please consider revising throughout the manuscript. Similarly, please consider replacing "rods" with "bars" (see for instance line 252).

2) Line 21: please define the acronym, all acronyms the first time they appear.

3) Introduction, lines 91-93: the authors should try to seek more balance between the literature review part and that describing the objectives and hence the innovation of the research.

4) Line 99: these values are taken from a reference. Please use some lines of text to describe how they have originally been obtained.

5) Figure 1: Please present/describe key features of the test setup.

6) Line 112: unclear, please rephrase.

7) Lines 117-119: the authors should elaborate this three lines long paragraph, justifying the assumed values and/or providing references for the assumptions made. Either elaborate or remove this case.

8) Section 3 tends to be rather long and quite difficult to follow. It would help if five to ten lines of text were added (perhaps at line 201) such that the readers are more naturally guided into the rationale (not necessarily just the numbers) that the authors are trying to convey. Needless to say that, this is the most important part of the revision work the authors should be prepared to undertake – I am pretty sure the manuscript will benefit.

9) Please state if the treatment is only analytical or not. Numerical, FEM-based simulations accompanying the results would help.

Author Response

Response to Reviewer 2 Comments

The authors sincerely thank the reviewer for their attention to the work, useful comments, as well as for the high assessment of our work.

Point 1: Lines 12-13: it is suggested to replace the expression “in the compressed and stretched zones” with “in zones under tension and compression”. Please consider revising throughout the manuscript. Similarly, please consider replacing “rods” with “bars” (see, for instance, line 252).

Response 1: Thank you very much for your careful examination of our paper. Following your valuable comment, we have checked many inaccuracies in the manuscript, and inaccurate descriptions such as “in the compressed and stretched zones” and “rods” instead of “bars”. We have corrected these inaccuracies throughout the article.

Point 2: Line 21: please define the acronym, all acronyms the first time they appear.

Response 2: We fixed it. The acronym for the stress-strain state is SSS.

Point 3: Introduction, lines 91-93: the authors should try to seek more balance between the literature review part and that describing the objectives and hence the innovation of the research.

Response 3: Thank you very much for your careful examination of our paper. We have reworked section 1 Introduction and describe the goals of our article more carefully. We noted that the bimodality of materials could have a critical impact on strength calculations, and therefore on the size and durability of structures. Dynamic effects and inertial forces, in combination with the bimodality, significantly change the SSS of structure.

Point 4: Line 99: these values are taken from a reference. Please use some lines of text to describe how they have originally been obtained.

Response 4: Thank you for your comment. The values of elastic modulus for tension and compression were taken in the work of S. Ambartsumyan. These data are obtained from experiments on uniaxial tension and compression

Point 5: Figure 1: Please present/describe key features of the test setup

Response 5: We have reworked section 2 Materials and methods and describe the device for experimental tests for the static three-point bend of a 4x20 mm beam. During the experiment, the deflection was measured at the point of application of force with hour-type indicator.

Point 6: Line 112: unclear, please, rephrase.

Response 6: We rework this paragraph.

For a steel beam, the theoretically obtained deflection of the beam, taking into account bimodularity, is 4% more than the experimental one and excluding bimodularity by 5.3%. For a silumin beam, the theoretical deflection is 0.7% more than the experimental one taking into account bimodularity and 5.4% without taking into account bimodularity. The results obtained confirm the need to take into account the bimodularity property when calculating the strength and stiffness of structural elements.

Point 7: Lines 117-119: the authors should elaborate this three lines long paragraph, justifying the assumed values and/or providing references for the assumptions made. Either elaborate or remove this case.

Response 7: Thank you for your attention. We rework this paragraph.

In the calculations of concrete structures, fiber-reinforced concrete with a tensile-compression modulus of 5000 MPa was used, if the material is considered as isotropic. Finding the heterogeneous properties of concrete, various elastic moduli under the tension of 5000 MPa and compression of 2550 MPa are accepted in the calculations.

Point 8: Section 3 tends to be rather long and quite difficult to follow. It would help if five to ten lines of text were added (perhaps at line 201) such that the readers are more naturally guided into the rationale (not necessarily just the numbers) that the authors are trying to convey. Needless to say that, this is the most critical part of the revision work, the authors should be prepared to undertake. I am pretty sure the manuscript will benefit.

Response 8: Thank you very much for your comment and careful examination of our paper. We rework section 3 and section 4. We remain four figures instead of 12 and describe these figures for easier understanding.

Point 9: Please state if the treatment is only analytical or not. Numerical, FEM-based simulations accompanying the results would help.

Response 9: Yes, it is only analytical. The main goal of the article is to obtain simple engineering expressions for calculating structures made of modular materials. We intentionally simplified the solution to avoid complex partial differential equations, which no one will solve in engineering practice. Numerical methods, such as the Finite Eelement Method, might be useful. I think we will be able to apply it to our next works.

Finally, we would like to thank you very much for your support!!!

Reviewer 3 Report

The authors present an interesting work entitled Transverse impact on rectangular metal and reinforced concrete beams considering bimodularity of the material.

The presented theme is not exactly innovative, however the paper in general is well organized and well structured having obtained interesting results.

In my opinion, the paper has conditions to be considered for publication, requiring only a few corrections, essentially formatting.

Author Response

The authors thank the reviewer for their attention to our work and appreciation of our efforts.

Round 2

Reviewer 1 Report

Nil

Reviewer 2 Report

The paper has been improved and may now be accepted for publication.